# Shade Modifies Behavioral and Physiological Responses of Low to Medium Production Dairy Cows at Pasture in an Integrated Crop-Livestock-Forest System

**DOI:** 10.3390/ani11082411

**Published:** 2021-08-15

**Authors:** Natani S. Reis, Isabel C. Ferreira, Lucas A. Mazocco, Ana Clara B. Souza, Gabriel A. S. Pinho, Álvaro M. da Fonseca Neto, Juaci V. Malaquias, Fernando A. Macena, Artur G. Muller, Carlos F. Martins, Luiz C. Balbino, Concepta M. McManus

**Affiliations:** 1Graduate Program in Veterinary Science, Campus Gloria, Federal University of Uberlandia, BR-050, Km 78, Uberlandia 38410-337, Brazil; natani.reis@ufu.br; 2Embrapa Cerrados, Br 020, Km 18, Planaltina 73310-970, Brazil; alvaro.neto@embrapa.br (Á.M.d.F.N.); juaci.malaquias@embrapa.br (J.V.M.); fernando.macena@embrapa.br (F.A.M.); artur.muller@embrapa.br (A.G.M.); carlos.martins@embrapa.br (C.F.M.); luizcarlos.balbino@embrapa.br (L.C.B.); 3Faculty of Agriculture and Veterinary Medicine, Campus Darcy Ribeiro, University of Brasília, Brasília 70910-900, Brazil; mazocco.lucas@gmail.com (L.A.M.); anacsouza.agro@gmail.com (A.C.B.S.); 4União Pioneira de Integração Social SEPS Q 712/912 Conj A-Asa Sul, Brasília 70390-125, Brazil; gabrielalberto.mv@gmail.com; 5Institute of Biology, Departmant of Physiological Sciences, Campus Darcy Ribeiro, University of Brasília, Brasília 70910-900, Brazil; concepta@unb.br

**Keywords:** ingestive behavior, dairy, heat stress, morphological characteristics, pasture

## Abstract

**Simple Summary:**

The husbandry of high-producing dairy cattle on pasture in tropical regions promotes heat stress and alters physiological and behavioral parameters. However, it is unknown how the physiological and behavioral responses of cows more adapted to tropical environments under heat stress conditions, with lower milk production under shaded pasture or full sun conditions. To respond to these questions, Gyr dairy and Girolando cows (5/8 Holstein 3/8 Gyr, and ½ Holstein ½ Gyr) were evaluated in full sun and under natural shade from trees. Behavioral, physiological, and morphological variables were measured, and it was concluded that shaded pasture for dairy zebu cows promotes animal welfare by mitigating animal surface temperature and increasing rumination time.

**Abstract:**

Under conditions of high temperature, humidity, and incidence of solar radiation, dairy cows use behavioral changes as a strategy to decrease the metabolic heat production at pasture. The objective of this study was to evaluate the behavioral and physiological responses of Gyr and Girolando (5/8 Holstein 3/8 Gyr, and ½ Holstein ½ Gyr) dairy cows submitted to environments with and without shade. The experiment was conducted at Embrapa Cerrados (Technology Center for Dairy Zebu breeds—CTZL), Brasilia, Distrito Federal—Brazil, with 48 Gyr and Girolando cows total in the lactation group, with low to medium milk production, in full sun or shade with *Eucalyptus urograndis* (267 plants/ha^−1^). The physiological and behavioral characteristics evaluated were panting score, superficial temperature, and time spent grazing, ruminating, and lying down. Other traits included skin and coat thickness, hair diameter, density and length, and predominant coat color. In addition, body measurements, such as body length, the height of withers, chest circumference, and shin circumference, were measured. Shaded cows had 34% longer rumination times than cows in full sun (*p* = 0.01). With a temperature-humidity index ranging from 79 to 83, the rumination time was 1.7 times higher in cows under shade (*p* = 0.01) during a 24-h period of observation. There were no significant differences in the grazing time between the environments, but lying time was 23% longer in cows under the sun (*p* = 0.01). The panting score was not influenced by the environment (*p* = 0.17). Girolando cows had a 35% higher panting score than Gyr cows (*p* = 0.01) regardless of the environment. The panting score increased two and a half times during the afternoon compared with the morning (*p* = 0.01). The surface and rectal core temperatures had significant differences between treatments and time of the day. Body measurements were not different between cows in both environments, but there was a difference between breeds. The use of trees in pastures with a silvopastoral system for dairy zebu cows is indicated to improve grazing behavior, as well as time spent ruminating and lying down.

## 1. Introduction

Dairy production in the Cerrado (savannah) biome in central Brazil is mainly at pasture. This region’s climate has a high potential for forage production. Still, high temperatures, humidity, and intense solar radiation affect the farmer’s ability to maintain balanced milk production throughout the year [1], as the climate affects the productivity and longevity of dairy cows [2,3].

Girolando animals are preferably adopted by all types of farmers (small to large) due to the productivity and rusticity of these animals when exposed to a tropical climate, especially in summer [4]. The Holstein and Gyr breeds represent almost 80% of the country’s milk production [1]. In environments with a higher incidence of radiation, high temperatures, and humidity, the combination of genetics and environment determines important parameters that benefit the production levels of the Girolando breed at pasture [5,6]. Dairy cows under heat stress conditions acquire metabolic heat from radiant energy in large quantities. Combined with the low heat exchange of the animal with the environment, this leads to an increase in body temperature, reduces dry matter intake and, consequently, lowers milk yields [7].

The use of trees in a pasture-based dairy system has been applied in many farms in the Cerrado region [8]. This integrated crop-livestock-forestry (ICLF) system leads to land use intensification, increasing productivity and profit. Product diversity, such as wood and resin production increases, thereby improving soil conservation, providing shade for animals, and nutrient dynamics for forage crops. In addition to the positive impact on the microclimate to improve thermal comfort for the animals, the use of eucalyptus trees contributes to the economic activity, with a sustainable system, producing animal by-products and wood products throughout the year, and with a future return, in line with the producer’s aims. In areas where the natural resources preservation, as well as the sustainable system, are under pressure, the demand to establish practices that offer alternatives of economic and social profitability are being studied and adopted in the livestock system [8,9].

High temperatures, humidity, and solar radiation incidence [10] are considered stress factors for dairy cattle. The use of natural or artificial shade in the pasture production system protects animals from intense solar radiation and modifies their radiation balance. Natural shading with tree species can reduce the incident radiant heat load on the animals by 30% [11]. In addition, shade use for dairy pasture production is an essential resource to increase the quantity and quality of oocytes and embryos produced in vitro [12].

However, the influence of the ambiance offered by tree shade in a silvopastoral system on dairy cows at lower milk yields needs to be investigated to understand the impact on the tropical production environment fully. It is hypothesized that zebu dairy cows in tropical pastures can experience heat stress, even if they present low to medium milk production. Natural shade is expected to favor the behavior of zebu dairy cows during grazing and lying down, and consequently, promote better welfare in tropical environments. Thus, the objective of this study was to evaluate the behavioral and physiological responses of Gyr and Girolando dairy cows submitted to grazing systems with shade in ICLF and full sun systems.

## 2. Material and Methods

The experiment was conducted at Embrapa Cerrados (Center of Technology for Dairy Zebu breeds—CTZL), Brasilia, DF, (15°57′09″ S, and 48°08′12″ W), the central region of Brazil, in the Cerrado biome, from February 2017 to February 2019.

### 2.1. Treatments and Experimental Area Description

Two environments (treatments) were used to examine heat stress in dairy cows in a total area of 16 hectares. Half the area (8 hectares) was the control treatment (animals kept at pasture in full sun), and the other half consisted of pasture-crop-forest integration (animals kept at pasture under the shade of *Eucalyptus urograndis*). Pasture in both cases was *Panicum maximum* cv. Mombaça. The 20 m-high eucalyptus trees, providing a shaded environment, were arranged in single rows with 1.5 m between trees and 25 m between rows, arranged approximately in the east-west direction, totaling 267 trees/ha with 8% tree cover area. The implementation of trees in this arrangement follows the technical recommendations of Embrapa 2019, taking into account the characteristics of the system as a whole, involving space for crops, animals, and machinery [13].

### 2.2. Animals and Experimental Design

The experiment was approved by the Ethics Committee on Animal Use of Embrapa Cerrados (CEUA/Embrapa Cerrados), Brasília—DF, Protocol No. 533-2541-1/2017. The experimental design was fully randomized with two treatments, the control in full sun and the shade under eucalyptus trees. Forty-eight Gyr and Girolando cows (24 of each breed) were used while in their lactation period during the two years of the experiment. Each cow was considered an experimental repetition. The average yield did not correct for the fat of Gyr and Girolando cows (5/8 Holstein 3/8 Gyr, and ½ Holstein ½ Gyr), and was 10 and 15 L.day^−1^, respectively.

### 2.3. Animal Measurements

Skin and coat thickness, as well as hair diameter, density, length, and color, were measured according to Silva’s method [14]. Skin (ST) and coat thickness (CT) were measured using an adipometer graduated in tenths of millimeters, taken from the upper region of the animal’s body close to the scapular region and the dorsum, flank, and hind leg. For hair count (HC), hair samples were collected from the upper central region of the scapula in an area of approximately 1 cm^2^ and collected with pliers to remove all hairs. The collected hairs were placed in paper envelopes and identified for later count and measurement of the ten longest hairs. The hairs were spread on a sheet of white paper for dark-colored hairs and a black surface for white hair samples, using tweezers and needles [14]. For hair length (HL), ten of the longest hairs were selected from each sample, measured with a ruler, and averaged.

The pigmentation of the skin, after trichotomy, and coat were evaluated by the CIELAB system using a colorimeter (Minolta^®^ model CR-10, Tokyo, Japan) once a year. The colorimeter, composed of a CIE photoresistor (Lab), detects the intensity of light reflected by the epidermis when the spotlight is directed at it. The L* values refer to luminosity (0 = black and 100 = white), the red-green component as the a* chroma (red color intensity) and the yellow-blue component the b* chroma (yellow color intensity) [15]. Three consecutive measurements were taken, and the average skin and coat were taken from 33 cows, 16 in full sun and 17 in shade, and these cows were 16 Gyr cows and 17 Girolando cows, during their lactation periods, dry or in-calf cows from the same lot were not measured.

Body measurements were taken from these same 33 animals with similar lactation phases. These included withers height (WH) as the highest point of the interscapular region, using a tape measure; body length (BL) from the tip of the pallet to the ischial tuberosity, using a hipometer; and shin (SC) and chest circumference (CC) with a tape measure [16,17].

The evaluations of animal behavior at pasture were made by direct visual observation, for 24 h uninterrupted, in 16 cows, 8 of each treatment (full sun and shade) and breed (4 Gyr and 4 Girolando), on eleven different dates in spring-summer and autumn seasons over two years. For this purpose, four trained observers kept watch. Two observers were assigned to each treatment and were placed to avoid interference in the cow’s behavior, with observation shifts of 6 h. Binoculars, chronometers, and flashlights were used. Each observer reported grazing time by the direct visual observation method [18]. The ethogram recorded the animal’s immediate activity, classifying grazing as the act of selecting grazing sites, bolus seized and handled; ruminating when the animal exhibits regurgitation and re-cordering of the bolus, the time between swallowing and regurgitation; and lying down when the animal is without activity or jaw movements. The observations were taken every 10 min, and the four animals of each genetic breed per treatment were evaluated during this period.

Thermography surface temperatures and panting scores were obtained from 34 cows on 17 dates from January to November 2017, with 18 animals under the full sun (8 Gyr and 10 Girolando) and 16 under shade (8 Gyr and 8 Girolando). Thermographic photos were obtained using an infrared camera T3000 series (FLIR^®^ Systems Inc., Wilsonville, OR, USA), with an emissivity coefficient of 0.98, temperature range (−20 to 400 °C) and accuracy of +/− 2%. For each animal, two photos were taken (lateral view of the whole body and udder) for each period: morning (7 a.m.) and afternoon (3 p.m.), at a distance of approximately two meters from the animal. The FLIR QuickReport^®^ v. 1.2 software was used for the data analyses of thermographic images in each region of the animal’s body (udder, croup, flank, eye, neck, and muzzle). A veterinary clinical thermometer was used to take the rectal temperature.

The panting score was measured in the morning and afternoon in their respective paddocks with natural shade or direct solar radiation and determined on a scale of zero to four, where zero means a normal breathing animal, 1: slightly increased, 2: moderate panting, 3: strong panting, and 4: severe panting [19].

### 2.4. Thermal Index Acquisition in Natural Shade and Full Sun

Climatic and microclimatic parameters (air temperature and humidity, dry and wet bulb temperatures, rainfall, wind speed) were obtained in both environments through weather stations with touchscreen display ITWH 1080 INSMART for the computation of thermal indexes. The black globe temperature was taken with a portable black globe thermometer ITWG2000 (INSTRUTEMP, Measuring Instruments Ltd., São Paulo, SP, Brazil). With a globe sensor from 0 °C to 80 °C, dry bulb sensor from 0 °C to 50 °C, relative humidity (RH) from 0% to 100% RH, with a resolution of 0.1 °C/0.1% RH and accuracy of 1 °C. These were placed in the shade of the ICLF treatment and in full sun. Measures were taken every hour from 7 a.m. to 5 p.m.

The temperature and humidity index (THI), calculated by the formula AT + 0.36 × TDP + 41.5 and the black globe temperature and humidity index (BGHI), calculated from the formula BGT + 0.36 × TDP + 41.5, were determined, with AT as the ambient temperature, BGT as the black globe temperature, and TDP as the dew point temperature [20,21].

THI was classified as normal (<74), alert (75–78), danger (79–83), and emergency (>84) to analyze its effect and interactions on behavioral response variables. The air temperature was classified into two categories (above and below 30 °C) for physiological reasons due to the dissipating of 80% of the latent body heat through evaporation from the skin [22].

### 2.5. Statistical Analysis

Behavioral variables were analyzed using a mixed model procedure, with the fixed effects of breed and treatment groups, the interactions (genetic breed *treatment), milk yield as a covariate, random effect of cow, and date of evaluation as a repeated measure in conditions of THI classification (normal, alert, danger, and emergency) and temperature classification (above and below 30 °C). Surface temperatures by thermography and rectal core temperatures were analyzed by PROC MIXED considering the fixed effect of treatment (full sun and shade), time of day (morning and afternoon), and the interactions of these factors, cow as a random effect, dates of sampling as a repeated measure in time for each season (rainy and dry), and milk yield as a covariate. The genetic breed was included in the model as a fixed effect, but without a significant effect, it was removed. The Bayesian-Schwarz criterion (BSC) was used to choose the best fits of the models.

Color, skin, coat thickness, and hair number data were analyzed using PROC GLM, considering the treatment and genetic breed effects and interactions. In all cases, the least squared means were compared by Tukey’s test at 5% probability.

Body measurements were analyzed considering the effect of treatment and genetic breed and tested for the interaction of the two factors by variance analysis. The F test was used to determine significance between factors.

Panting scores were tested by Kruskal–Wallis Test with *p* < 5% considered significant, considering separate effects of the treatment (full sun and shade), genetic breed (Gyr and Girolando), season (dry and rainy), and day period (morning and afternoon).

The normality of the behavioral and physiological data was tested by the Shapiro–Wilk test. All data presented a normal distribution.

All statistical analyses were carried out in SAS v 9.4 (Statistical Analysis System Institute, Cary, NC, USA).

## 3. Results

The maximum BGHI obtained in the study (88.9) was in the dry season during the afternoon period in the full sun treatment. There were no significant interactions between the time of day and treatment for environmental variables (*p* = 0.63). THI values in the rainy season showed an effect for time of day (*p* = 0.01), with higher values in the afternoon (78). In the dry season, there was an effect of the shade and time of day (*p* = 0.01) (Table 1).

The THI was mitigated with shade by 2.5% to 5.2%. In the dry season, trees reduced the ambient temperature (AT) by 2.5% in the afternoon and 7.6% in the morning (*p* = 0.01). In the rainy season, the AT changed both in the morning and afternoon (*p* = 0.01). The time of day and treatment affected the black globe temperature (BGT) in the dry season, reducing it by 14.7% in the afternoon (*p* = 0.01). The BGT did not vary due to treatment (*p* = 0.05) and time of day (*p* = 0.27) in the rainy season, probably because the weather had a higher cloud cover and had more mist (Table 1).

The hair number was unaffected by breed (*p* = 0.70) and environment (*p* = 0.80). The hair of Girolando cows was longer (0.14 mm) than Gyr cows (*p* = 0.01). However, length did not differ between treatments (*p* = 0.36). Skin (*p* = 0.78) and coat (*p* = 0.56) color were not different between environments. Gyr coat color (a = red color intensity) was 2.2 times higher than Girolando (*p* = 0.01) (Table 2).

Body measurements (wither height, chest circumference, body length, and shin circumference) did not differ between cows in the full sun and shade ICLF treatments but showed differences between breeds (Table 3).

Girolando were longer, taller, and had a higher chest circumference than Gyr cows. Gyr cows exposed to full sun showed higher coat thickness (0.27 cm) than Gyr cows in the shade ICLF (Table 4).

Cows under shade spent 29% to 34% more time ruminating compared to those in full sun at <30 °C and >30 °C, respectively. When the THI was at danger levels, the rumination time of cows under shade was 1.7 times higher than those in the sun. There were no significant differences between treatments for grazing times, but time spent lying down was 19% longer in the ICLF (Table 5) than in the sun.

The evaluation of the effect of genetic breed and interaction with THI classes and treatment on rumination had no significant differences (*p* = 0.29; *p* = 0.94 and *p* = 0.21, respectively). These effects on idleness were only from genetic breed (*p* = 0.02; Gyr = 506 min and Girolando = 470 min), the interaction with THI classes and treatment had *p* = 0.12 and *p* = 0.35, respectively. The same response pattern was observed for intake (genetic breed *p* = 0.01; Gyr = 483 min and Girolando = 530 min; interaction with THI classes *p* = 0.18 and interaction with treatment *p* = 0.13).

The response pattern was similar when rating temperature below and above 30 °C. There was no effect of interactions on rumination, idleness, and ingestion. Gyr cows spent more time in idleness (521 min) than Girolando cows (470 min) (*p* = 0.01). The Girolando cows spent more time in the pasture ingestion (531 min) than the Gyr cows (477 min) (*p* = 0.01).

Shade interfered positively with cows’ rumination when the temperature was classified below and above 30 °C and in THI, indicating alert and dangerous situations. Both at normal and dangerous THI levels, cows in the full sun spent longer lying down (Table 5).

Surface temperatures at different parts of the body were significantly lower in shaded cows, and there was also a time-of-day effect, with lower temperatures in the morning. Cows in full sun and the afternoon had higher surface temperatures. The rectal temperature of cows was higher in the afternoon, in full sun (0.7 °C) and ICLF (0.6 °C) (Table 6).

ICLF and season did not affect the cow’s panting score. Girolando cows had a 35% higher panting score than Gyr cows, independent of the treatment. The panting score increased two and a half times in the afternoon compared to the morning period (Table 7).

## 4. Discussion

### 4.1. Thermal Index Obtained in Natural Shade and Full Sun

The maximum BGHI (88.9) that occurs in the dry season during the afternoon period in full sun (Table 1) is considered an emergency [21]. Most THI values were in the range of 79 to 84, indicating danger. Such conditions above the thermoneutral zone suggest that animals probably suffer heat stress. Despite high THI values, tree shade helped to mitigate this rate by 7.3%. Similar results were reported in studies where the presence of shade in the pasture mitigated THI values by 3.7% [22], and under THI conditions at 72, the yield of milk and feed intake started to decline. When THI values reached 76 or higher, the decrease in milk yield is clearly reduced [23].

In the rainy season, THI values were influenced only by the time of day (morning and afternoon), and in the dry season, THI changed with the time of day and treatment (presence of shade) (Table 1). THI in both periods was above 72, indicating stressful conditions for dairy cows [24]. THI was lower in the morning shaded treatments but even so was considered critical, while in the afternoon, with sun exposure, it reached a level considered dangerous [25]. THI was mitigated by shade by 2.5% to 5.2%. Trees in dry seasons reduced ambient temperatures (AT) in the afternoon by 2.5% and in the morning by 7.6%. These values indicate that even under heat stress conditions, with a THI above 72, the shade density of the trees present in the shade environment helps to reduce the thermal discomfort by at least 4 °C in the dry period of the year. Similar results were found in which an environment shaded by eucalyptus in an ILPF system, THI values are reduced by up to 2.7% when compared to full sun environments [26]. Even in shaded conditions, THI values throughout the year were above expected for thermal comfort (72), reaching values of 75 during the hottest hours of the day and up to 81 in full sun [26].

During the rainy season, the AT changed in both morning and afternoon periods (Table 1).

### 4.2. Morphological Characteristics of Skin, Coat and Body Measures of Cows in Shaded ICLF and Full Sun Systems

Determination of coat pigmentation and skin color, according to the colorimeter readings, presents a color scale with a value of 0 for black and 100 for white color according to Muller’s method [15]. Skin and coat color were not different between the two environments. The Gyr coat colors were 2.2 times lighter than Girolando (Table 2). Crossbred cows, with zebu genetic composition, tend to present a more pigmented epidermis with lighter hair as a result of natural selection, increasing protection of deeper tissues from short-waves ultraviolet radiation, which crosses the thin layer of the coat easily [27].

Animals that present a thinner coat possess accelerated excess body heat dissipation via radiation. As this radiation passes through the coat, it is detained by the melanin granule layer of the epidermis, evidencing a favorable selection of light coat and dark skin characteristics in addition to physical structure and quantity of hairs by area unit of the coat [28,29].

Gyr cows exposed to full sun presented 32% thicker coats (Table 4) than Gyr cows in the ICLF. Some studies have shown that long hair can be a heat insulator, serving as a buffer between the environment and the animal’s body [30,31]. An important point to note here is that, in the dry season, although the temperature is high during the day, it can fall well below 10 °C at night [32,33], as skies in this region at this time of year do not have cloud cover, so a lot of heat is lost from the earth’s surface by radiation. While Girolando cows contain *Bos taurus taurus* genes, and therefore may show a higher resistance to these colder temperatures, Gyr (*Bos taurus indicus)* do not have an adaptation to low temperatures [34]. This may have stimulated coat growth in the unshaded pastures, while the cows in the shaded areas would not have undergone the same stress.

However, observations in more animals are needed to verify how the environment can affect coat thickness in each season. These results suggest a difficulty for Gyr cows to lose heat, and the coat cover should be as thin as possible, with short, thick, and well-seated hairs to facilitate latent and sensible heat loss [6,10,35,36].

Some characteristics, such as shorter and less dense hair in crossbred cows, with lower values during the summer, are considered appropriate for optimizing evaporative and convective thermolysis, based on high temperatures and humidity during summer when higher heat stress is observed [37]. An adequate coat, therefore, provides physiological adjustments to heat stress, such as losing heat through vasodilation and faster sweating into the atmosphere, since the thicker the coat and the longer the hair, the more thermal insulation the animal will have, consequently affecting heat dissipation [10].

Several coat characteristics are preferable for cows in tropical climates, such as short, high-density coats, high diameter, and light-colored hair with an inclination less than 40°, and pigmented skin [38,39]. These characteristics allow more protection from solar radiation and heat stress, contributing to animal comfort and better production under field conditions [40].

### 4.3. Animal Behavior and Ingestion

Cows in the shade spent 29% to 34% more time ruminating than in full sun (Table 5), indicating that those in full sun tend to spend more idle time. In a study conducted in Brazil, significant differences in time lying down, rumination, and looking for shade in dairy cows in different pasture systems were found [7]. Higher rumination and lying times were found in cows with available shade. Cows in the full sun spent most of their time close to water sources, with fewer rumination times, especially on days with higher THI during the afternoon, as a strategy to increase heat loss by both convection and conduction while standing or lying down on a moist, cold surface [41].

Beef cattle under shade in a silvopastoral system were seen to change their ingestive behavior during warmer months of the year. Also time spent resting or lying down was longer for animals under sun exposure in hotter periods of the year with temperatures above 29 °C, in an attempt to reduce the excess metabolic heat produced [42].

### 4.4. Surface and Rectal Temperatures

The internal rectal temperature of cows under full sun and shade remained within the physiologically normal range. The normal value was considered to be up to 38.5 °C [43]. The surface temperature measured at different parts of the body remained within the normal range. These responses suggest that cows under the full sun could adapt to environmental challenges to perform heat loss to maintain thermal balance. As the cows in the sun had more idle time and reduced rumination, this reduces their metabolic heat output [10]. However, the higher rectal temperature of the cows in the sun indicates that they were unable to dissipate body heat, even without reaching the physiological threshold for heat stress. This indicates that the reduced metabolic heat production was not sufficient to overcome increased stress caused by environmental factors, especially at the hotter times of the day.

### 4.5. Panting Score

The panting score has been determined as a good indicator of stress in cattle [44,45]. The shade and season did not affect the cow’s panting score. Girolando cows had a slightly increased panting score than Gyr cows, regardless of environment. In the afternoon, the panting score was two and a half times higher than in the morning period (Table 7). One of the main mechanisms in heat dissipation is the panting behavior in dairy cows, and this mechanism has a negative correlation with wind speed [19]. The increased rate of respiration is indicative that these animals are losing heat in an attempt to maintain homeothermy. In a study conducted in São Paulo—Brazil, with Girolando animals, summer and winter seasons affected the gland activities responsible for sweating and increased panting scores to control the body temperature of those cows during summer. Under heat stress conditions, those animals lose water through increased sweating and panting, leading to dehydration and reduction in production performance [23].

Girolando cows presented better adaptive responses by activating a heat loss mechanism, through increased panting, to high temperatures in the dry season compared to Gyr cows [19]. Although the number of hairs was not statistically different between breeds, the lighter coat color, pigmented skin, shorter hair and thinner cover, and larger body measurements were observed in the Girolando breed, indicating better ability to dissipate excess body heat during the experimental period [39,45]. Larger animals have a lower metabolic rate than smaller ones [46], and animals with longer, thinner appendages show increased heat loss [47].

When in environments shaded by trees, animals spent more time grazing and ruminating than animals in full sun. While in the shade, there was no difference between time spent lying down above and below 30 °C, in the sun, animals tended (not significantly) to spend less time lying under the higher temperatures. This would be to increase sensible water loss, increase body surface area exposed to air movement or convection, and dissipate heat to the environment. Thus, medium to low production Gyr and Girolando cows tolerate heat but show behavioral changes to maintain homeothermy.

The presence of tree shade in the pasture of an integrated production system for zebu dairy cows was found to promote the cow’s well-being by increasing rumination and reducing their skin temperature. However, it did not affect behavioral parameters, such as panting score and grazing times. This may be due to the adaptation of these breeds to these environment, since crossbred Gyr (with Holstein-Friesian and Simental) cows were more productive in the Cerrado system than purebred Gyr for calf and milk production, as well as reproductive traits [48], and while genetic group and environment (sun and shade) did not affect rectal temperature, respiration rate was increased in crossbred animals [2].

## 5. Conclusions

Environmental parameters indicate heat stress conditions for dairy cows in all seasons, with the most challenging conditions in the afternoon and during the dry season for the Cerrado biome. The use of trees in pastures of a silvopastoral system for dairy zebu cows is recommended to improve animal behavior parameters, such as ingestive behavior, rumination, and time spent lying down, as well as to improve cow’s welfare in the shade group by providing shelter from hot sunny environments.

## Figures and Tables

**Table 1 animals-11-02411-t001:** Black globe humidity index (BGHI), temperature and humidity index (THI), air temperature (TA), and black globe (BG) temperature (°C) obtained from the full sun and shade environment with Integrated Crop-Livestock-Forestry (ICLF) in the morning and afternoon, during the dry and rainy seasons.

Parameter	Season	Morning	Afternoon			Significance		
		Full Sun	Morning Shade	Full Sun	Afternoon Shade	Treatment	Time of Day	Treat × Time
BGHI	Dry	81.4 ± 1.2	76.4 ± 1.4	88.9 ± 1.1	82.4 ± 1.0	0.01	0.01	0.63
	Rainy	82.1 ± 2.3	79.6 ± 1.1	85.9 ± 1.7	80.3 ± 1.9	0.04	0.23	0.41
THI	Dry	76.0 ± 0.8	72.0 ± 0.8	82.0 ± 0.9	80.0 ± 0,8	0.01	0.01	0.17
	Rainy	73.8 ± 1.3	72.3 ± 1.3	78.0 ± 1.4	75.5 ± 1.4	0.15	0.01	0.70
AT	Dry	26.8 ± 0.8	24.0 ± 0.7	31.5 ± 0.9	30.7 ± 0.7	0.03	0.01	0.19
	Rainy	24.5 ± 0.9	23.4 ± 0.9	28.6 ± 1.0	26.6 ± 1.0	0.14	0.01	0.65
BG	Dry	32.1 ± 1.1	27.9 ± 1.1	38.6 ± 1.1	32.9 ± 1.1	0.01	0.01	0.51
	Rainy	31.7 ± 1.5	29.7 ± 1.6	35.0 ± 1.8	30.2 ± 1.6	0.05	0.27	0.42

ICLF—integrated crop–livestock–forestry. Morning = 7 a.m. Afternoon = 3 p.m.

**Table 2 animals-11-02411-t002:** Average length of the 10 longest hairs, number of hairs per cm^2^, and skin and coat color (Silva, 2000) of Gyr and Girolando cows in full sun and ICLF shade production systems.

Parameter	Full Sun	Shade	*p*-Value *	Gyr	Girolando	*p*-Value *
Number of hairs (hair/cm^2^)	805 ± 108	795 ± 153	0.8	791 ± 67	809 ± 175	0.70
Size 10 > hair (mm)	0.53 ± 0.10	0.60 ± 0.20	0.36	0.49 ± 0.05	0.63 ± 0.20	0.01
Skin color	7.8	7.1	0.78	9.6	5.3	0.09
Fur color	13.8	12.6	0.56	18.4	8.3	0.01

* Tukey–Kramer test at 5% probability; ICLF—integrated crop–livestock–forestry; *p*—level of significance.

**Table 3 animals-11-02411-t003:** Average skin thickness, body length, wither height, shin, and thoracic length according to the genetic group of cows.

Parameter	Gyr	Girolando	*p*-Value *
Skin thickness (cm)	0.62	0.61	0.83
Body length (m)	1.42	1.63	0.01
Wither height (m)	1.29	1.40	0.01
Shin circumference (cm)	21.4	20.9	0.68
Thoracic circumference (cm)	1.76	2.01	0.01

* Tukey–Kramer test at 5% probability.

**Table 4 animals-11-02411-t004:** Mean coat thickness (cm) according to treatments full sun and ICLF with shade and genetic breed.

Treatment	Genetic Breed	
	Gyr	Girolando
Coat thickness in full sun (cm)	1.09 ^Aa^	0.91 ^Aa^
Coat thickness in shade (cm)	0.82 ^Ba^	0.95 ^Aa^

^A, B^ Different capital letters in the columns indicate statistical difference by Tukey test at 5% significance; ^a^ Different lowercase letters in the rows indicate statistical difference by Tukey test at 5% significance; ICLF—integrated crop–livestock–forestry.

**Table 5 animals-11-02411-t005:** Mean time, in minutes, of ingestive behavior during 24 h, of dairy cows under full sun and shade ICLF and under different categories of air temperature and THI.

Ingestive Behavior	Air Temperature	Full Sun	Shade	*p*-Value
Grazing	<30 °C	484.8 ± 14.6	506.6 ± 14.9	0.72
	>30 °C	518.7 ± 19.5	520.1 ± 19.5	0.99
*p*-Value		0.5155	0.9496	
Rumination	<30 °C	301.0 ± 13.0	404.1 ± 13.7	0.01
	>30 °C	312.5 ± 16.4	404.4 ± 16.8	0.01
*p*-Value		0.9403	1.000	
Lying down	<30 °C	561.0 ± 13.6	456.0 ± 14.1	0.01
	>30 °C	520.5 ± 17.8	455.6 ± 18.0	0.05
*p*-Value		0.2712	1.000	
	THI Classes			
Grazing	Normal	492.8 ± 19.3 ^A^	502.5 ± 14.6 ^A^	0.99
	Alert	486.2 ± 22.3 ^A^	545.6 ± 27.3 ^A^	0.54
	Danger	509.6 ± 27.3 ^A^	510.0 ± 27.3 ^A^	0.99
Rumination	Normal	269.0 ± 14.0 ^B^	384.2 ± 10.6 ^B^	0.01
	Alert	388.7 ± 16.2 ^A^	424.3 ± 19.8 ^AB^	0.73
	Danger	269.6 ± 14.0 ^B^	468.7 ± 19.9 ^A^	0.01
Lying down	Normal	561.8 ± 16.3 ^A^	479.8 ± 12.3 ^A^	0.01
	Alert	477.5 ± 18.9 ^B^	402.5 ± 23.1 ^B^	0.13
	Danger	585.3 ± 16.3 ^A^	424.3 ± 23.0 ^AB^	0.01

^A, B^ Distinct capital letters in the column differ by the Tukey Kramer test at 5% significance level; ICLF—integrated crop–livestock–forestry; THI—temperature-humidity index.

**Table 6 animals-11-02411-t006:** Average surface temperature by thermography and rectal temperature in Gyr and Girolando cows in full sun and shade ICLF in the morning and afternoon in different body sites.

Region	Morning		Afternoon		Significance *		
	Full Sun	Shade	Full Sun	Shade	Treatment	Period	Treat × Per
Udder	32.5 ± 0.1	31.6 ± 0.2	35.8 ± 0.1	34.6 ± 0.2	0.01	0.01	0.30
Croup	32.2 ± 0.3	31.7 ± 0.3	35.4 ± 0.3	34.0 ± 0.3	0.01	0.01	0.18
Flank	32.8 ± 0.2	32.3 ± 0.2	35.7 ± 0.2	34.8 ± 0.2	0.01	0.01	0.52
Neck	32.3 ± 0.2	32.1 ± 0.2	35.6 ± 0.2	34.3 ± 0.2	0.01	0.01	0.04
Eye	32.8 ± 0.2	32.4 ± 0.2	35.8 ± 0.2	34.6 ± 0.2	0.01	0.01	0.08
Muzzle	29.8 ± 0.3	28.7 ± 0.3	33.4 ± 0.3	31.7 ± 0.3	0.01	0.01	0.36
Rectum	37.5 ± 0.07	37.9 ± 0.0	38.2 ± 0.0	38.5 ± 0.0	0.01	0.01	0,83

* Tukey–Kramer test; ICLF—integrated crop–livestock–forestry.

**Table 7 animals-11-02411-t007:** Panting scores of Gyr and Girolando dairy cows in full sun and shade ICLF obtained in the dry and rainy seasons in the morning and afternoon periods.

	Mean Panting Score			
Environment	Full Sun	Shade ICLF	s.e.	Significance *
	0.92	0.84	0.04	0.16
Genetic Breed	Gyr	Girolando		
Season	0.75	1.00	0.04	0.01
	Rainy	Dry		
Period	0.97	0.81	0.04	0.03
	Morning	Afternoon		
	0.51	1.28	0.03	0.01

* Non-parametric Kruskal–Wallis test. s.e.—standard error; ICLF—integrated crop–livestock–forestry.

## Data Availability

The authors declare that the data presented in this study are available on request from the corresponding author.

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
