# Peer review of "Shade Modifies Behavioral and Physiological Responses of Low to Medium Production Dairy Cows at Pasture in an Integrated Crop-Livestock-Forest System"

_animals, 2021, doi:10.3390/ani11082411_

Round 1

Reviewer 1 Report

The article is very interesting and necessary. It brings new scientific knowledge. However, References must be numbered in order of appearance in the text and references list must be written according to the style used in Animal (MDPI). It is also necessary to correct mistakes on lines 17, 363, and 402 in the text. In line of 462, write down the remaining authors.

Author Response

Dear Reviewer, the replies to your suggestions follow,

Point 1: The article is very interesting and necessary. It brings new scientific knowledge. However, References must be numbered in order of appearance in the text and references list must be written according to the style used in Animal (MDPI).

Response 1: The References have been numbered according to the order in which they appear in the citations and have been adjusted to the style used by Animal (MDPI).

Point 2: It is also necessary to correct mistakes on lines 17, 363, and 402 in the text. In line of 462, write down the remaining authors.

Response 2: the mistakes made in lines, 17, 363 and 402 have been corrected according to the suggestion. In the line 462, each author has been described with their functions in separate lines.

Reviewer 2 Report

This study stated the role of shade in behavioral and physiological responses of low to medium production dairy cows. The explanations are not clear enough, there are some major and minor concerns to be addresses.

Major concerns;
1. It seems that the author did not write the manuscript according to the journal's author guidelines. For example, there should be a “simple summary” before the abstract. In addition, the expression of references is wrong, please revise it refer to the requirements of the journal. The “implications” seems to be superfluous. You should mark the subtitle with an ordinal number. Tables should be interspersed in the results section. The conclusion is missing at the end of the text.

2. The discussion is a bit bulky. A large part of the discussion is merely a description of the results, without explanation. You should combine more references to explain the results. Line 322-323 “Gyr cows exposed to full sun showed 32% thicker coats (Table 4) than Gyr cows in the ICLF” Authors need to explain it in the discussion section.

3. Line 367-368 “As the cows in the sun had more idle time and reduced rumination, this reduces their metabolic heat output (McManus et al., 2020)”. However, the rectal temperature of cows in the sun was higher than that in the shade (table 6). How do you explain it?

Minor concerns;

1. You have to revise the grammar of your sentences, e.g. Line 72, “demonstrate” ‎»“demonstrated”, Line 75 “state” ‎»“stated”, Line 81…

2. Line 93 the reference was inserted in the wrong place.

3. Line 104 It seems that this hypothesis did not agree with your study. Independently ?

4. Line 106 improve the behavior? You should define it further.

5. Line 184 7:00 a.m. and 3:00 p.m.

6. Line 201 7:00 a.m. to 5:00 p.m.

7. Line 209-211 Please add reference(s).

8. Line 270-283 P-value should be presented in this section.

9. Line 290-291 states ? starts?

10. Line 292-293 the decrease in milk yield is severely reduced ?

11. Line 293-294 Please add reference(s).

12. Line 295-301 It's hard for readers to understand what you mean.

13. Line 309 It should be showed as a reference.

14. Line362-363 This sentence is very vague.

15. Line 395-400 The positive impact of eucalyptus is a little redundant here, and it is not from the research content.

16. The experimental animals are “low to medium production dairy cows”. It is better for authors to supplement production performance data, which would Increase the productive significance of the study. This is just a suggestion.

17. Line 468 References The author must modify the format of the references according to the author's guidelines.

18. Line 590 Table4. Full sun (cm)? Shade (cm)?

19. The format of the table must be carefully modified to meet the requirements of the journal.

Author Response

Dear Reviewer, the replies to your suggestions follow,

Major concerns

Point 1: It seems that the author did not write the manuscript according to the journal's author guidelines. For example, there should be a “simple summary” before the abstract. In addition, the expression of references is wrong, please revise it refer to the requirements of the journal. The “implications” seems to be superfluous. You should mark the subtitle with an ordinal number. Tables should be interspersed in the results section. The conclusion is missing at the end of the text.

Response 1: The paper was corrected according to the style of the journal. Summary was placed before the Abstract. The References were corrected according to the journal's style. Implications were taken out of context, since it was superficial, and the last article published by the journal did not present Implications. The Subtitles were enumerated as suggested, the Tables were adequate in the style of the journal, and the Conclusion was inserted at the end of the text as suggested.

Point 2: The discussion is a bit bulky. A large part of the discussion is merely a description of the results, without explanation. You should combine more references to explain the results. Line 322-323 “Gyr cows exposed to full sun showed 32% thicker coats (Table 4) than Gyr cows in the ICLF” Authors need to explain it in the discussion section.

Response 2: The explanations of the results have been added along with their references, the sentence in line 322-323 about the coat thickness of Gyr cows has been adjusted according to the suggestion, data that contribute to the understanding of the results have been presented, followed by their respective references.

Point 3: Line 367-368 “As the cows in the sun had more idle time and reduced rumination, this reduces their metabolic heat output (McManus et al., 2020)”. However, the rectal temperature of cows in the sun was higher than that in the shade (table 6). How do you explain it?

Response 3: The full sentence was corrected according to suggestion, in response to the questioning, it was presented in the text that the reduced metabolic heat production was not sufficient to overcome increased stress caused by environmental factors, especially at the hotter times of day. 

Minor concerns;

Point 4: You have to revise the grammar of your sentences, e.g., Line 72, “demonstrate” ‎» “demonstrated”, Line 75 “state” ‎» “stated”, Line 81…

Response 4: Some words were deleted when the sentence was rewritten to fit the suggestions, others were corrected according to the guidelines.

Point 5: Line 93 the reference was inserted in the wrong place.

Response 5: Has been accepted and corrected.

Point 6: Line 104 It seems that this hypothesis did not agree with your study. Independently?

Response 6: Has been accepted and corrected.

Point 7: Line 106 improve the behavior? You should define it further.

Response 7: Writing error, ‘favors’ is more appropriate to the context.

Point 8: Line 184 7:00 a.m. and 3:00 p.m.

Response 8: Has been accepted and corrected.

Point 9: Line 201 7:00 a.m. to 5:00 p.m.

Response 9: Has been accepted and corrected.

Point 10: Line 209-211 Please add reference(s).

Response 10: The references has been added.

Point 11: Line 270-283 P-value should be presented in this section.

Response 11: P-value has been inserted in the corresponding sentences.

Point 12: Line 290-291 states? starts? (foi reescrito para adaptar ao formato das referências e estas palavras foram deletadas)

Response 12: Has been accepted and corrected.

Point 13: Line 292-293 the decrease in milk yield is severely reduced?

Response 13: Writing error, a more appropriate word has been added to the context.

Point 14: Line 293-294 Please add reference(s).

Response 14: The references has been added.

Point 15: Line 295-301 It's hard for readers to understand what you mean.

Response 15: The sentence has been rewritten for easier reading.

Point 16: Line 309 It should be showed as a reference.

Response 16: Has been accepted and corrected.

Point 17: Line362-363 This sentence is very vague.

Response 17: The sentence has been rewritten for better comprehension and added some references.

Point 18: Line 395-400 The positive impact of eucalyptus is a little redundant here, and it is not from the research content.

Response 18: Has been accepted and corrected (deleted).

Point 19: The experimental animals are “low to medium production dairy cows”. It is better for authors to supplement production performance data, which would Increase the productive significance of the study. This is just a suggestion.

Response 19: This suggestion was not taken.

Point 20: Line 468 References The author must modify the format of the references according to the author's guidelines.

Response 20: Has been adequate according to the style of the journal.

Point 21: Line 590 Table4. Full sun (cm)? Shade (cm)?

Response 21: Has been accepted and corrected.

Point 22: The format of the table must be carefully modified to meet the requirements of the journal

Response 22: Has been adequate according to the style of the journal.

Reviewer 3 Report

The study "Shade modifies behavioural and physiological responses of low to medium production dairy cows at 4 pasture in an Integrated Crop-Livestock-Forest system" measured the behaviour-production responses to different environments using two breed types. 

Overall, the study has major implications for the welfare and production of dairy cows reared in warm temperatures using behaviour responses to identify signs of distress.  

The manuscript requires edits and clarification in several areas listed above:

Abstract:

Please clarify if is coat color in line 33

Not clear what avaliation means in line 38

When referring to rectal temperature, add rectal “core” temperature (line 44)

Line 36, 38, 40, 42, 44. Write P values within 2 decimals consistently.

Materials and methods

In lines 133-135. You stated that Gyr and Girolando cows (5/8 Hostein 3/8 Gyr and ½ Holstein ½ Gyr) were producing 10 and 15 kg/day, respectively. Did you use 2 types of Girolando cross if so they were evenly mixed in the treatments since you only mentioned 2 breeds previously? 

Line 157-164. What was the reason for using only 33 animals to collect the body and coat measurements (please explain)? 

Line 166. Inconsistent American English spelling. Please use eighter “behaviour” or “behavior” consistently in the paper.

Line 168. Describe what genetic group is.

Line 171. Behavior

Line 166-177. Please describe the visual observation schedule (day time of observations) 

Line 178-188. Please describe what type of camera was used, company, accuracy, temperature range etc.

Line 214 Behavioral

Line 215. I recommend you used Breed group instead of genetic to avoid confusions

Line 214-225. Why the breed was not considered as a fixed variable for your infrared measures?

Line 226. Inconsistent English spelling (colour)

Line 213-239. Did your behaviour data comply with the normality assumptions?

Results Section 

Line 241-283. Add P-values to the text to back up your results.

Discussion section

Line 378-381. Change “performance reduction” to “reduction in production performance”

Line 403-406. Increased rumination, idle time, and surface temperature are not considered welfare aspects, however, those signs can identify a reduction in the well-being of an animal. Rewrite the sentence to “The presence of tree shade in the pasture of an integrated production system for zebu dairy cows was found to promote the cow’s well-being by increasing rumination and reducing their skin temperature.”

Line 406-407. Panting score and grazing times are not physiological parameters. Those are behavioral parameters. 

Line 414-417. Again rumination, ingestion and laying are not welfare parameters, those are behavior parameters. Welfare parameters can be access to shelter, nutrition, good health, absents of pain and suffer, etc. So you can conclude that cow’s welfare was improved in the shade group by providing shelter to hot sunny environments. 

Line 403. Scientifically you can’t partially confirm a hypothesis either you accepted (null hypothesis) or reject it (alternative hypothesis). I recommend reformulating your hypothesis because you found interesting results. 

Tables in General. Keep your data and P values within 2 decimals consistently

Table 6. Add Infrared behind thermography 

Author Response

Dear Reviewer, the replies to your suggestions follow,

Abstract

Point 1: Please clarify if is coat color in line 33

Response 1: Yes, it is.

Point 2: Not clear what avaliation means in line 38

Response 2: Writing error, a more appropriate word has been added to the context.

Point 3: When referring to rectal temperature, add rectal “core” temperature (line 44)

Response 3: Has been accepted and corrected.

Point 4: Line 36, 38, 40, 42, 44. Write P values within 2 decimals consistently.

Response 4: P-value has been inserted in the corresponding sentences.

Materials and methods

Point 5: In lines 133-135. You stated that Gyr and Girolando cows (5/8 Hostein 3/8 Gyr and ½ Holstein ½ Gyr) were producing 10 and 15 kg/day, respectively. Did you use 2 types of Girolando cross if so, they were evenly mixed in the treatments since you only mentioned 2 breeds previously?

Response 5: Yes. They were evenly mixed, it has been corrected in the sentence.

Point 6: Line 157-164. What was the reason for using only 33 animals to collect the body and coat measurements (please explain)?

Response 6: Animals with a more uniform lactation phase were evaluated, the sentence was adequate for better understanding.

Point 7: Line 166. Inconsistent American English spelling. Please use eighter “behaviour” or “behavior” consistently in the paper.

Response 7: Has been accepted and corrected to behavior in all the text.

Point 8: Line 168. Describe what genetic group is.

Response 8: Has been changed to ‘breed’ for better understanding.

Point 9: Line 171. Behavior

Response 9: Has been accepted and corrected.

Point 10: Line 166-177. Please describe the visual observation schedule (day time of observations)

Response 10: We do not have the information about the exact days of observations, it has been added to the text the most used methodologies recognized in the literature.

Point 11: Line 178-188. Please describe what type of camera was used, company, accuracy, temperature range etc.

Response 11: Has been described in the text as suggested.

Point 12: Line 214 Behavioral

Response 12: Has been accepted and corrected.

Point 13: Line 215. I recommend you used Breed group instead of genetic to avoid confusions

Response 13: Has been accepted and corrected.

Point 14: Line 214-225. Why the breed was not considered as a fixed variable for your infrared measures?

Response 14: Race was included in the model, but without significant effect, it was removed later. This information was inserted in the text for better understanding.

Point 15: Line 226. Inconsistent English spelling (colour)

Response 15: Has been accepted and corrected.

Point 16: Line 213-239. Did your behaviour data comply with the normality assumptions?

Response 16: The normality of the behavioral and physiological data was tested by the Shapiro-Wilk test. All data showed normal distribution. This information was inserted in the text for better understanding.

Results Section

Point 17: Line 241-283. Add P-values to the text to back up your results.

Response 17: P-value has been inserted in the corresponding sentences.

Discussion section

Point 18: Line 378-381. Change “performance reduction” to “reduction in production performance”

Response 18: Has been accepted and corrected.

Point 19: Line 403-406. Increased rumination, idle time, and surface temperature are not considered welfare aspects; however, those signs can identify a reduction in the well-being of an animal. Rewrite the sentence to “The presence of tree shade in the pasture of an integrated production system for zebu dairy cows was found to promote the cow’s well-being by increasing rumination and reducing their skin temperature.”

Response 19: Has been accepted and corrected.

Point 20: Line 406-407. Panting score and grazing times are not physiological parameters. Those are behavioral parameters.

Response 20: Has been accepted and corrected.

Point 21: Line 414-417. Again rumination, ingestion and laying are not welfare parameters, those are behavior parameters. Welfare parameters can be access to shelter, nutrition, good health, absents of pain and suffer, etc. So, you can conclude that cow’s welfare was improved in the shade group by providing shelter to hot sunny environments.

Response 21: Has been accepted and corrected.

Point 22: Line 403. Scientifically you can’t partially confirm a hypothesis either you accepted (null hypothesis) or reject it (alternative hypothesis). I recommend reformulating your hypothesis because you found interesting results.

Response 22: Has been accepted and the Hypothesis has been reformulated to better understanding.

Point 23: Tables in General. Keep your data and P values within 2 decimals consistently

Response 23: Has been adjusted to the style of the journal.

Point 24: Table 6. Add Infrared behind thermography

Response 24: Has been accepted and corrected.

Reviewer 4 Report

This manuscript reports a field experiment comparing behavioural and some physiological responses of Gur and Girolando dairy cattle grazing pastures with and without shade, with the former offered as a model for an Integrated Crop-Livestock-Forestry system.  The paper is generally very clearly written, with Results reported and discussed clearly.  Please especially consider the points relating to statements made in the Discussion (indicated with ‘’), which do not appear to be well-supported by the results reported in the manuscript.  Other comments here have been made for the authors to consider as potential mild improvements to the manuscript.  Thank you for the opportunity to review this manuscript.

Line 116-125:    It would be helpful to know whether this arrangement and density of trees is typical of ICLF systems in the Cerrado region.

Line 135:           Please confirm this is 10 and 15 kg.day-1 of milk solids?

Line 218-19:      Please explain in the Statistical Analysis section why temperature classification was included as a binomial variable in the statistical model – was this from exploratory statistical analysis or is there a biological reason for this modelling approach? If the latter, can a reference be supplied?

Line 262-4:       Please clarify what the 32% refers to – is it that Gyr-full sun cows had 32% thicker skin than Gyr-shade cows?

Lines 267-71:    Is the sentence “The mean time spent grazing didn't change comparing full sun and ICLF treatments” (lines 270-1) a repeat of the sentence “There were no significant differences between treatments for grazing times” (line 267-8)? If so, please remove one.

Line 301-4:       For readers without detailed knowledge of the meaning of particular THI scores, it would be helpful if the authors provided more comment about the practical importance of decreases in THI in ICLF of 2.5-5.2% (line 301-2) and 7.6% (line 303)

Line 346:           Correct “29 a 34%” to “29 to 34%”

Lines 372-3:   The comment about Girolando cows having 35% greater panting score seems to be over-emphasised, since both breeds had mean panting scores ≤ 1 (slightly increased) – please would the authors consider clarifying the importance/clinical significance of this observation more fully.

Line 382-:       The statement “Girolando cows presented better adaptive responses to high temperatures…” does not seem consistent with the observation and comments in the preceding paragraph, which state that Girolando cows had 35% greater panting score than Gyr cows, as greater panting score is considered an indicator of heat stress. Please clarify this apparent inconsistency.

Lines 382-87:    Please justify the assertion that the different body measurements observed in Girolando cows convey better tolerance of high temperatures by citing appropriate references.

Lines 387-9:   The statement “When in environments shaded by Eucalyptus trees, these animals [Girolando] had longer periods of grazing and rumination than animals in full sun.” does not appear to be supported by evidence presented in the Results. For example, there is no statistically significant interaction between breed and treatment for ingestive behaviour times presented in Table 5 or in the Results text (I would expect them to be presented around lines 264-74).  Could this please be clarified by the authors. 

Lines 394-402:            These assertions are not justified by this experiment or the results presented in the manuscript and should be moved to the Introduction or deleted.

Table 5:             Please consider re-wording the row label “pasture intake” to something like “grazing”, which is the term used to describe the categorisation of cow behaviour in the Method.

                        I suggest Table 5 would be slightly clearer to read if the sub-heading “Air Temperature” were moved down one row below the main column headings, so that it has similar formatting to the other sub-heading in the table, “THI classes”

Table 6:            Should “Ubber” be “udder”?

Author Response

Dear Reviewer, the replies to your suggestions follow,

Point 1: Line 116-125: It would be helpful to know whether this arrangement and density of trees is typical of ICLF systems in the Cerrado region.

Response 1: Information has been added to the sentence that supports the arrangement used according to the literature.

Point 2: Line 135:  Please confirm this is 10 and 15 kg.day-1 of milk solids?

Response 2: The production values were not corrected for fat, i.e., milk production without correction for 4% fat. this information was inserted in the text for better understanding.

Point 3: Line 218-19:      Please explain in the Statistical Analysis section why temperature classification was included as a binomial variable in the statistical model – was this from exploratory statistical analysis or is there a biological reason for this modelling approach? If the latter, can a reference be supplied?

Response 3: This is not an exploratory analysis and there is a biological reason. Information grounded in literature was inserted for better understanding.

Point 4: Line 262-4:       Please clarify what the 32% refers to – is it that Gyr-full sun cows had 32% thicker skin than Gyr-shade cows?

Response 4: The sentence has been rewritten for better understanding.

Point 5: Lines 267-71:    Is the sentence “The mean time spent grazing didn't change comparing full sun and ICLF treatments” (lines 270-1) a repeat of the sentence “There were no significant differences between treatments for grazing times” (line 267-8)? If so, please remove one.

Response 5: The sentence has been rewritten for better understanding (delete the information repeated).

Point 6: Line 301-4:       For readers without detailed knowledge of the meaning of particular THI scores, it would be helpful if the authors provided more comment about the practical importance of decreases in THI in ICLF of 2.5-5.2% (line 301-2) and 7.6% (line 303)

Response 6: Information has been added to the sentence that supports the data used according to the literature.

Point 7: Line 346:  Correct “29 a 34%” to “29 to 34%”

Response 7: Has been accepted and corrected.

Point 8: † Lines 372-3:   The comment about Girolando cows having 35% greater panting score seems to be over-emphasized, since both breeds had mean panting scores ≤ 1 (slightly increased) – please would the authors consider clarifying the importance/clinical significance of this observation more fully.

Response 8: When the animals increase their breathing rate it indicates that they are losing heat to maintain homeothermy. Information and references were inserted in the sentence for better understanding.

Point 9: † Line 382-:       The statement “Girolando cows presented better adaptive responses to high temperatures…” does not seem consistent with the observation and comments in the preceding paragraph, which state that Girolando cows had 35% greater panting score than Gyr cows, as greater panting score is considered an indicator of heat stress. Please clarify this apparent inconsistency.

Response 9: When animals increase panting, they indicate that they are triggering heat loss mechanism. Information and references were inserted in the sentence for better understanding.

Point 10: Lines 382-87:    Please justify the assertion that the different body measurements observed in Girolando cows convey better tolerance of high temperatures by citing appropriate references.

Response 10: Literature-based information has been inserted for a better understanding of the text.

Point 11: † Lines 387-9:   The statement “When in environments shaded by Eucalyptus trees, these animals [Girolando] had longer periods of grazing and rumination than animals in full sun.” does not appear to be supported by evidence presented in the Results. For example, there is no statistically significant interaction between breed and treatment for ingestive behaviour times presented in Table 5 or in the Results text (I would expect them to be presented around lines 264-74).  Could this please be clarified by the authors.

Response 11: While in the shade there was no difference between time spent lying down above and below 30oC, in the sun animals tended (not significant) to spend less time lying under the higher temperatures. This would be to increase sensible water loss, increase body surface area exposed to air movement or convection and dissipate heat to the environment.

Point 12: † Lines 394-402:  These assertions are not justified by this experiment or the results presented in the manuscript and should be moved to the Introduction or deleted.

Response 12: Has been moved to introduction according to suggestion.

Point 13: Table 5:   Please consider re-wording the row label “pasture intake” to something like “grazing”, which is the term used to describe the categorization of cow behavior in the Method.

Response 13: Has been accepted and corrected.

Point 14: I suggest Table 5 would be slightly clearer to read if the sub-heading “Air Temperature” were moved down one row below the main column headings, so that it has similar formatting to the other sub-heading in the table, “THI classes”

Response 14: Has been accepted and corrected.

Point 15: Table 6: Should “Ubber” be “udder”?

Response 15: Has been accepted and corrected.

Round 2

Reviewer 2 Report

The authors should further revise the manuscript according to the journal's author guidelines before publication.